# Dyslipidemia and Meibomian Gland Dysfunction: Utility of Lipidomics and Experimental Prospects with a Diet-Induced Obesity Mouse Model

**DOI:** 10.3390/ijms20143505

**Published:** 2019-07-17

**Authors:** Eugene A. Osae, Philipp Steven, Rachel Redfern, Samuel Hanlon, C. Wayne Smith, Rolando E. Rumbaut, Alan R. Burns

**Affiliations:** 1University of Houston, College of Optometry, Houston, TX 77204, USA; 2Department of Ophthalmology, Division for Dry-Eye and Ocular GvHD, Medical Faculty, University of Cologne, 50937 Cologne, Germany; 3Baylor College of Medicine, Children’s Nutrition Center, Houston, TX 77030, USA; 4Center for Translational Research on Inflammatory Diseases (CTRID), Michael E. DeBakey Veterans Affairs Medical Center, Houston, TX 77030, USA

**Keywords:** dyslipidemia, meibomian gland dysfunction, lipidomics, ocular surface, dry eye

## Abstract

Meibomian gland dysfunction (MGD) is the leading cause of dry eye disease and loss of ocular surface homeostasis. Increasingly, several observational clinical studies suggest that dyslipidemia (elevated blood cholesterol, triglyceride or lipoprotein levels) can initiate the development of MGD. However, conclusive evidence is lacking, and an experimental approach using a suitable model is necessary to interrogate the relationship between dyslipidemia and MGD. This systematic review discusses current knowledge on the associations between dyslipidemia and MGD. We briefly introduce a diet-induced obesity model where mice develop dyslipidemia, which can serve as a potential tool for investigating the effects of dyslipidemia on the meibomian gland. Finally, the utility of lipidomics to examine the link between dyslipidemia and MGD is considered.

## 1. Introduction

The meibomian gland (MG) is a modified sebaceous gland in the eyelids which produces meibum, the lipid component of the tear film [1]. Meibum is essential for retarding tear film evaporation, thereby preventing dry eye [1]. Meibum has the ability to lower surface tension, thus aiding in the spreading of the tear film at the ocular surface [2]. Additionally, it possesses some antimicrobial properties believed to prevent ocular surface infections [3].

Regulation of the MG is complex and thought to involve autonomic nervous control and hormonal influences [1] (Figure 1). The MG is the only sebaceous gland in the body that is rich in sensory, sympathetic, and parasympathetic innervation. It has been shown to express neurotransmitters and neuropeptides including serotonin, calcitonin gene-related peptide, neurotensin, somatostatin, neuropeptide-Y and γ-aminobutyric acid. It also expresses mRNAs for cholinergic, dopaminergic, glutaminergic and adrenergic receptors [1,4,5]. However, it is unclear how these neurochemicals are released around the MG and how they exert their physiological effects on the MG. Regarding hormonal control of the MG, a few studies suggest that androgenic hormones and insulin may be necessary for the growth and normal functioning of the MG [1,2,3,6].

As a contributor to the precorneal tear film, the MG is an integral component of the lacrimal functional unit and the ocular surface; together with the cornea, the conjunctiva, main and accessory lacrimal glands (Figure 1) [7]. Pathology of the MG, clinically termed meibomian gland dysfunction (MGD), is the leading cause of dry eye disease [8,9]. MGD is known to be present in up to 70 percent of dry eye cases seen globally, and in the United States alone, MGD affects over 7 million people [1,10].

The international workshop on meibomian gland dysfunction defined MGD as a chronic, diffuse abnormality of the MG and may be characterized by terminal duct obstruction and quantitative and qualitative changes in the glandular secretion [1,11]. These can result in alterations of the tear film, symptoms of eye irritation, clinically apparent inflammation and ocular surface disease [1]. MGD exists in two broad categories: as low delivery and high delivery forms [1]. Low delivery MGD is further divided into hyposecretory and obstructive types, where reduced volume of meibum secretion results from glandular dropout and/or blockage of MG orifices, respectively [12,13]. Higher delivery or hypersecretory forms of MGD are characterized by a large volume of meibum at the lid margin on gentle application of pressure on the tarsus. Hypersecretory MGD is often associated with seborrheic dermatitis, hence it is also referred to as seborrheic MGD [12,13].

This subtype of MGD has been linked to acne rosacea, where skin sebocytes produce excess sebum. Since the MG is a modified sebaceous gland, the proposed explanation is that the seborrhea may be due to an end-organ hyper-response of the MG to androgens [1,14]. On one hand, reduction in levels of some androgens like testosterone, dehydroepiandrosterone and its conjugated dehydroepiandrosterone sulfate are considered as potential risk factors for developing MGD [15]. Other well-known risk factors include desiccative stress at the ocular surface due to low humidity environments [16]. To date, the etiology of MGD is not fully understood, but aging, prolonged contact lens wear, recurrent lid infections, use of certain drugs (e.g., isotretinoin, chemotherapy) and irradiation are known risk factors for developing MGD [17,18,19,20,21].

Dyslipidemia has also been linked to the development of MGD, but direct evidence supporting this relationship is lacking [22,23,24,25]. Dyslipidemia is a disorder of systemic lipid metabolism which is characterized by (abnormal) increased levels of total blood cholesterol (TC), triglycerides (TGs), low-density lipoproteins (LDL) and/or a reduction in high-density lipoproteins (HDL). It is a major component of the metabolic syndrome which includes abdominal obesity, systemic inflammation, insulin resistance, hypertension and hyperglycemia [26,27].

Dyslipidemia is a major risk factor for heart disease, but exactly how it might contribute to MGD is unknown [28,29,30,31]. Plasma lipids have been suggested to affect the MG as well, but the evidence is largely circumstantial [22,23,24,31,32]. Moreover, dating back to 1957, one can find reports stating that there is no relationship between the status of plasma lipids and MG lipid composition [1,22,23,24,30,31,32]. The idea that plasma lipids can affect the MG is anecdotal and likely based on the fact that it is a lipid-synthesizing organ and alterations in systemic lipid metabolism could affect its structure and function [22,23,24,31,32]. Whether pathologic levels of systemic lipids contribute to MGD remains speculative [1,33]. It is thus important to determine if experimental evidence will support a link between dyslipidemia and MGD. This takes on added significance given the global epidemic status of the metabolic syndrome—which affects approximately 25% of the adult population and shows increasing incidence in children [26,27,28].

The pathology of MGD is complex and poorly understood. Linking dyslipidemia to the pathological mechanism driving MGD would fill a critical gap in our understanding. Adopting a lipidomics approach, it should be possible to determine if a particular plasma lipid profile in dyslipidemia causes qualitative (compositional) and quantitative changes in meibum [34,35,36]. Since MGD is characterized by structural changes in the MG [1], it is necessary to adopt careful structural and ultrastructural examination of the MG and its lipids (meibum) synthesizing apparatus (e.g., MG peroxisomes) through powerful imaging techniques like meibography and electron microscopy [37,38]. These approaches may be reinforced by the application of coherent anti-Stroked Raman spectroscopy (CARS) to probe MG lipidome (and proteome) in the presence of dyslipidemia [16,39]. CARS has the advantage of informing if MG lipid profiles change during dyslipidemia by producing readouts of vibrational signatures of chemical bonds inherent in the different MG species. This, when combined with combined immunofluorescence, allows us to obtain vital MG structural information through its spectral imaging capabilities [16,39].

To do this, a suitable animal model is required. Herein, we review current literature on dyslipidemia and its associations with MGD. A diet-induced obesity model, where mice fed a high-fat diet develop dyslipidemia, is briefly introduced as a potentially useful model to experimentally study the link between dyslipidemia and MGD [40]. Finally, we conclude with some experimental prospects, placing emphasis on how lipidomics can help us understand the relationship between dyslipidemia and MGD [41].

## 2. Methods

### 2.1. Literature Search Strategy

We systematically searched for publications related to meibomian gland dysfunction (MGD) and dyslipidemia from January 2004 to May 2019 and limited to studies published in English. Five databases—Medline (via the Google Scholar search engine), Scopus, the Cochrane Library, the Center for Agriculture and Bioscience International (CABI) (via the CAB Direct electronic platform) and the World Health Organization Library Information System (WHOLIS)—were searched. We applied a modified sensitive search method with search terms adapted to find publications on MGD and dyslipidemia. Since dyslipidemia encompasses various lipid profiles, including increased blood levels of cholesterol, triglycerides or lipoproteins (i.e., high- and low-density lipoproteins), we varied the strings of search text used; for example, “dyslipidemia and meibomian gland dysfunction”, “hypercholesterolemia and meibomian gland dysfunction”. Citations and full-text papers were exported to Endnote X6 citation manager. The search strategy and results from the listed databases are summarized in a flowchart (Figure 2).

### 2.2. Inclusion and Exclusion Criteria

Studies qualified for inclusion if they:(a)Were original research papers published in English(b)Involved human subjects(c)Described study design(d)Contained quantitative information on the clinically accepted definition and/or diagnosis of dyslipidemia and meibomian gland dysfunction(e)Attempted to determine if there is a relationship between dyslipidemia and meibomian gland dysfunction

Studies were excluded if they were performed on animals, contained outcomes that were solely qualitative or had characteristics that contradicted the above inclusion criteria.

### 2.3. Appraisal of Included Studies

Due to differences in study design, methodology and the levels of evidence presented per study, we attempted to rate each study that met our inclusion criteria using a modified Newcastle–Ottawa Scale (NOS) for case-control and cohort studies [42,43]. The aim of this evaluation algorithm is to assign a numerical score to the overall quality of each study. This assessment is done with consideration of study bias, sample characteristics, imprecision, inconsistencies, and indirectness in measuring outcomes of interest. Attention is also paid to effect size and residual confounders.

### 2.4. Data Extraction and Analysis

We captured information on the country of study and/or ethnicity of study participants, study design/type and number of participants per study. We also captured information on participants’ age, diagnostic criteria for meibomian gland dysfunction and dyslipidemia. Prevalence ratios were computed from prevalence values of dyslipidemia for case and control groups to aid in making inferences. Major outcomes are also summarized and descriptively presented. We acknowledge that while it would have been useful to add a meta-analysis to this review, it was not possible because of the differences in methods of data acquisition and presentation formats across studies.

## 3. Results

We obtained 16 publications from our search, but only five studies that were properly indexed met the inclusion criteria [22,23,24,31,32]. The remainder (one animal study, one review paper and nine duplicates) were excluded (Figure 2). Diagnoses of meibomian gland dysfunction (MGD) and dyslipidemia were common to the studies included. Across these studies, diagnosis of MGD was based on slit lamp examination for glandular obstruction or expressibility and quality (i.e., physical appearance) of meibum. None of the studies provide detailed information on the assessment of, or document, MG loss or atrophy, such as through the use of meibography. Detailed information on how the investigators diagnosed and/or staged MGD is presented in Appendix A. To determine if subjects had clinically meaningful MGD, only one study performed dry eye symptom assessment [31]. Dyslipidemia was diagnosed by the presence of one or more of the following: fasting serum levels of TC > 200 mg/dL, TGs > 150 mg/dL, LDL > 130 mg/dL or HDL < 40 mg/dL [22,23,24,31,32]. A summary of findings from the studies are presented in order of the most recent to the oldest publication.

The most recent study relevant to this topic was published in 2018 by Guliani et al. [31]. This was a prospective observational study conducted over an 18-month period. The aim was to correlate the severity of MGD with serum lipoprotein levels. The study involved 90 cases of MGD and 90 age- and sex-matched controls. The age range of participants was 18–54 years. The study reports a strong association between increasing severity of MGD and LDL levels > 130 mg/dL, *p* < 0.001.

In addition to these findings, the study reports that subjects with advanced stage MGD tended to show serum TGs levels > 150 mg/dL and TC levels > 200 mg/dL. Further, severity of MGD was significantly associated with age and being female. It is worth noting that severe MGD cases were accompanied by surprisingly healthy levels of HDL (i.e., >40 mg/dL). Compared to LDLs, HDLs are generally deemed beneficial to health [41]. It is also expected that increasing levels of LDLs should be accompanied by a reduction in levels of HDLs (i.e., <40 mg/dL) [44]. Therefore, with respect to the speculated relationship between MGD and dyslipidemia, the expectation is that pathologically high levels of LDLs, TGs, TC and decreased (<40 mg/dL) HDL level should negatively affect the MG. Given that HDL is protective, the observed levels of HDL should rather be beneficial to the MG, and thus it is unclear why MGD patients showed a rather healthy level of HDLs (>40 mg/dL) in this study [24,45].

The second most recent investigation into the relationship between dyslipidemia and MGD is a retrospective case-control study conducted by Braich et al. [22], in 2016. The investigators sought to determine if there was an association between dyslipidemia and MGD. The study involved 109 cases of MGD (age range: 20–72 years) and 115 controls (age range: 19–75 years). All participants had no known history of dyslipidemia, so they were screened for dyslipidemia as previously described [31] and associations were determined between MGD and HDL, LDL, total cholesterol and triglyceride levels. Study participants were excluded if they were below 18 years of age or had a history of hypercholesterolemia. Further, participants did not qualify for the study if they were pregnant, diabetic or had a history of Sjögren’s syndrome or ocular surgery (9 months prior to the study). Other exclusion criteria included use of lipid-lowering drugs, omega-3 fatty acids, tear-altering substances and steroids (< 4 weeks prior to the study).

The major finding was that MGD patients showed about three times greater prevalence of dyslipidemia (64%) compare to controls (18%). The odds of having dyslipidemia were 18 times greater among the MGD patients compared to controls (*p* < 0.01). Specifically, patients with TG levels > 150 mg/dL were three times more likely to have MGD (*p* < 0.03). Those with LDL > 130 mg/dL were nine times more likely to have MGD, whereas those with TC > 200 mg/dL showed about a 14 times greater chance of having MGD. In addition to these, old age (> 65 years) and being male were found to significantly raise the odds (four to six times more) of having dyslipidemia, *p* < 0.01.

In 2013, Pinna et al. [32] published an observational case-control study to investigate a possible correlation between MGD and hypercholesterolemia. The study involved young and middle-aged participants aged 14–58 years. These consisted of 60 patients with moderate to severe symptoms of MGD and 63 controls who were accompanying friends or hospital staff. The defined exclusion criteria were similar to those of Braich et al. [22], and qualifying participants were also expected to be 18–54 years old [32]. The participants did not have an established dyslipidemic status, so fasting blood lipid levels were assessed as described previously [22,31]. Data from this study show that the prevalence of hypercholesterolemia was significantly higher among the MGD patients (58.3%) than controls (6.3%), *p* < 0.0001. Additionally, mean levels of TC and LDL were significantly higher among cases compared to controls, *p* < 0.0001. The investigators reported that MGD was significantly associated with increased levels of TC and LDL (Odds ratio (OR), 1.07; 95% confidence interval, 1.04–1.09; *p* < 0.001). While mean HDL levels were lower among cases, they were significantly associated with MGD (OR, 1.11; 95% CI, 1.06–1.17, *p* < 0.001).

Prior to the study by Pinna et al. [32], another study relevant to this topic was published in 2013 by Bukhari [23]. This was a prospective cohort study involving 236 participants (132 MGD cases and 104 controls) with an average age of 49.4 years. The aim was to examine the relationship between dyslipidemia and severity of MGD. Diagnoses of dyslipidemia and MGD as well as exclusion criteria were similar to that described in the other studies [31,32]. The major finding was that the overall prevalence of dyslipidemia was essentially identical between MGD patients (52.3%) and controls (53.8%). Unlike the other studies described, there were no significant correlations between MGD and dyslipidemia [22,31,32].

Lastly, the earliest study in our review that investigated the potential link between dyslipidemia and MGD was published by in 2010 by Dao et al. [24]. This was also a retrospective case-control study to determine if there was an association between MGD and dyslipidemia. It involved 46 subjects (after exclusion) with moderate to severe MGD and an average age of 52.2 years. Cases were compared to historical controls (average age 46.6 years) obtained from the National Health and Nutrition Examination Survey (NHANES). Diagnosis for dyslipidemia and MGD and exclusion criteria were similar to the other studies described above [22,23,31,32]. This study also reported a significantly higher prevalence of dyslipidemia, specifically hypercholesterolemia, among MGD cases (67.4%) compared to historical controls (45.1%), *p* = 0.0012. Similar to the study by Giuliani et al. [31] increased HDL (> 40 mg/dL) contributed to hypercholesterolemia. Interestingly, MGD cases showed significantly less prevalence of hypertriglyceridemia (15.2%) compared to controls (33.1%), *p* = 0.0049. The levels of LDL were no different for case and control groups.

Table 1 and Table 2 separately summarize important findings and NOS evaluation for each of the studies included.

## 4. Discussion

### 4.1. Blood Lipid Profiles and MGD; Study Limitations and Confounders

The relationship between plasma lipid status and meibomian gland health remains unclear and warrants investigation [1,30]. Therefore, the role of dyslipidemia in the development of MGD remains a grey area [1,25,45]. In spite of this, these studies provide useful insights into the potential effects of dysregulated systemic lipid metabolism on the MG, with shared similarities in their approach to examine whether a relationship exists between MGD and dyslipidemia [22,23,24,29,30].

The most relevant observation across all but one of the studies was the finding that patients with MGD had a high prevalence of dyslipidemia relative to controls [22,24,31,32]. This was contrasted by results in one of the studies where prevalence of dyslipidemia among patients with MGD was no different from those without MGD [22]. In fact, for this particular study, the prevalence ratio for dyslipidemia among cases and controls was ~1.0 (Table 1), leading the authors to speculate that dyslipidemia might not be the sole driving factor for the development of MGD [23].

While this was the case for this individual study [22], the other studies reported significant associations between elevated levels of blood TCs, TGs, LDLs, and decreased levels of HDL [22,31,32]. Further, two of the studies reported a significant association between MGD and elevated (> 40 mg/dL) blood levels of HDL [31,32]. Similarly, in one of the studies where specific tests of association between elevated HDL and MGD were not performed, the investigators concluded that elevated HDL may have a unique role in the development of MGD [24]. This was based on the observation that study participants with moderate to severe MGD showed increased levels of HDL [24]. What draws our attention is that in many metabolism-related studies, elevated HDL levels tend to be beneficial and usually not associated with pathologic states. This is because HDL, also known as “good cholesterol”, works to clear systemic circulation of “bad cholesterol” (LDL), thus reducing the risk for cardiovascular disease [46,47,48,49]. Whether high levels of HDL are selectively detrimental to the MG is yet to be explored [24,25,32].

Further, MGD is a multifactorial condition, making it challenging to conclude that the findings from these studies are largely attributable to dyslipidemia. Ageing, prolonged contact lens wear, and adverse environmental conditions including desiccating stress and the use of certain drugs like isotretinoin are known to negatively affect the MG [1,16,18]. Other factors that can influence the development of MGD include autoimmune conditions like Sjögren’s syndrome, ocular graft-versus-host diseases and, rosacea and hormonal insufficiencies [50,51,52].

Collectively, these studies underscore the need to better understand the relationship between dyslipidemia and MGD. Even though some of these studies report significant associations between MGD, age and sex of participants [22,23,24,31], little or no information is provided about how these variables may interact with other risk factors in the development of MGD. For example, only Pinna et al. [32] commented on the contact lens wear status of a few of the subjects; however, it was uncertain how this observation could impact their findings. Further, it has widely been published that ageing affects the MG by decreasing meibocyte differentiation, cell cycling and impaired peroxisome proliferator-activated receptor (PPAR)–γ signaling, leading to reduced secretory activity [53]. On its own, ageing is a significant risk factor for impaired metabolism, hence dyslipidemia [24,54,55]. Therefore, in drawing conclusions about the link between MGD and dyslipidemia, it is critical to apply an analytical model which considers the complex interactions between, e.g., age, sex and dyslipidemia and how these could separately or synergistically affect the MG.

Overall, while these studies provide useful information about the potential involvement of dyslipidemia in the pathogenesis of MGD, they carry several limitations. The studies are observational in nature with similar clinical settings, thus making it impossible to establish a “cause and effect” relationship [22,23,24,31,32]. Moreover, all five studies admitted a challenge with the sample characteristics. In the study by Dao et al. [24] for instance, historical controls obtained from the NHANES were slightly younger (mean age = 52.2 years) than actual cases studies (mean age = 46.6 years). According to the investigators, this could have contributed to the observed increase in the prevalence of hypercholesterolemia among the cases compared to the controls, given that old age is a risk factor for both MGD and impaired lipid metabolism [1,24].

Some of the studies acknowledge limitations with sample size, recommending larger studies to definitively probe if any relationship exists between MGD and dyslipidemia [22,24,31,32]. However, considering the fact that the majority of these studies were restricted to homogeneous groups of participants of solely Indian or Italian ethnicity, future larger population-based studies should aim at including participants of diverse ethnic backgrounds [22,31,32]. This will be very important in strengthening the observed associations and the generalizability of conclusions.

To date, only these few studies have come close in their attempts to investigate the relationship between dyslipidemia and MGD [22,23,24,31,32]. They provide some useful information about the potential link between MGD and dyslipidemia, but the considerable challenges with their study designs merit further prospective studies, especially at the basic science level [25]. This makes it useful to study this relationship in an experimental animal model, where dyslipidemia could be induced and controlled and the corresponding effects on the MG examined through lipidomics and imaging.

Consequently, the next sub-sections of this review are focused, albeit briefly, on discussing experimental prospects regarding this subject. A diet-induced obesity model (DIO), where mice fed a high-fat diet develop dyslipidemia, is introduced, and the power of lipidomics to enhance our understanding of the relationship between dyslipidemia and MGD is considered [56,57].

### 4.2. The Diet-Induced Obesity Mouse Model

Our lab uses the DIO (metabolic syndrome) mouse model to study the effects of a high-fat diet (HFD) on corneal nerve morphology and function as well as wound healing. We believe that this is an excellent model to determine if dyslipidemia is linked to MGD. All animal protocols are approved by the Institutional Animal Care and use Committee of the University of Houston and/or Baylor College of Medicine. Male C57/BL6 mice are usually fed a high-fat diet (42% kcal milk fat, diet #112734, Dyet Inc, Bethlehem, PA) for 5, 10 and 15 weeks [40]. Control mice are fed normal mouse chow for the same period. By 5 weeks, the mice fed an HFD develop dyslipidemia, characterized by an atherogenic profile of increased blood triglycerides and cholesteryl esters. Unlike the mice fed normal chow, we have unpublished observations of “milky tear fluid” at the lid margins suggesting changes in the meibomian gland which we suspect at this time to be consistent with the hypersecretory form of MGD [1].

The mice fed an HFD develop dyslipidemia in a natural fashion that is analogous to humans consuming the Western diet and share similarities in meibomian gland organization (Figure 3) as well as meibum composition [35]. Several studies have reported meibum composition in humans [35,57,58] and other species, and Butovich et al. [35] have shown that human and mouse meibum samples are similar in terms of percentage composition of their wax esters, cholesterol, cholesteryl esters, triglycerides, dihydrolanosterol, dihydrolanosterol esters, diacylated diols, *O*-acyl-ω-hydroxy fatty acids, squalene and ceramides [35]. These similarities make the DIO mouse model an excellent tool for evaluating the possibility of a link between dyslipidemia and MGD.

### 4.3. Application of Lipidomics to Investigate the Link between Dyslipidemia and MGD in the Experimental Situation

The term “lipidomics” is derived from the word “lipidome”, which refers to the complete lipid profile within cells, tissues or an organism [59]. The lipidome is major subset of a living system’s (i.e., cell or organism) metabolome, together with other major biomolecules such as proteins, sugars or nucleic acids [60]. Lipidomics encompasses the study of pathways and networks of cellular lipids in biological systems. This is achieved through quantitative and comprehensive definition of the lipidomes of biological sources at the smallest organelle level to whole organs [61,62].

Even though it is a relatively recent branch of metabolomics, no other “omics” discipline has experienced the tremendous growth of lipidomics in its development and application [62]. The field of lipidomics has rapidly advanced to include technologies such as mass spectrometry, nuclear magnetic resonance spectroscopy, fluorescence spectroscopy, interferometry and computational methods [59,61]. These techniques are used in conjunction with bioinformatics tools for comprehensive profiling of lipids [63].

Lipidomics has been very crucial in our understanding of the role of lipids in health and disease [41]. It has been instrumental in providing mechanistic insights into disturbances in cellular or systemic lipid metabolism, trafficking and homeostasis, and how these disturbances can lead to health conditions like obesity, diabetes, atherosclerosis, hypertension, systemic inflammation, some cancers and neurological disorders [41,64,65,66,67]. Information on the use of lipidomics in understanding eye disease is, however, limited. For example, only a handful of studies have applied lipidomics to elucidate biochemical properties of tear film lipids (meibum) and the specific aberrations in tear fluid which can lead to dry eye disease and some retinal diseases [41,68,69,70].

Butovich et al. have made significant contributions to our understanding of MG lipids (meibum): their biochemistry and functional properties at the ocular surface using the power of lipidomics [35,36,57,68,71,72]. In particular, pioneering works from this group and other labs provide extensive information about meibum composition and the structural detail of specific lipids [34,35,36,55,56,68,71]. Further, some of their works provide insight into the similarities in meibum across species [35]. According to Butovich et al., human and murine meibum share close similarities in their makeup. They show comparable relative abundances in meibum wax esters, cholesterol, cholesteryl esters, triglycerides sphingomyelins, sterols, sterol esters and acylated omega-hydroxyl fatty acids [35]. The list is inexhaustive given that analyses of meibum across species is still expanding. For example, the development of lipid standards for the characterization of certain meibum components is still ongoing [73].

Given the above information, meibum composition is definitely very complex [34,35]. In fact, the biochemistry of meibum is as intricate as its biosynthesis [1,35]. The current understanding is that meibum lipids could be synthesized de novo within the MG acini or taken up from the bloodstream. However, compared to the “uptake from bloodstream” theory, the de novo synthesis theory has received more support because enzymes involved in meibum synthesis including transestersases have been detected in MG acinar cells [1,6]. There is currently no evidence that lipids are taken from the bloodstream for incorporation into meibum lipids, except sex steroidal hormones which are lipid in nature with certain specific growth and functional effects on the MG [6,30,50]. It is possible that the uptake of these hormones could change with changing diet or plasma lipid status, such as that which occurs in dyslipidemia, and consequently affect MG health, hence MGD.

Moreover, even though there is scarce evidence on the uptake of other plasma lipids by the MG, it is not impossible that the MG could take up and incorporate lipids into the meibum from the bloodstream. This is because the MG is surrounded by a rich vascular network of blood vessels in the outer eyelid skin and palpebral conjunctiva, and we have unpublished observations of the close relationship between tarsal blood vessels and MG acini [1] (Figure 4). Blood and meibum lipid profiles share certain commonalities including the presence of triglycerides, cholesterol, cholesterol esters, sphingomyelins, and ceramides [34,74,75]. Thus, it is hypothetically possible that meibum components could be taken up from the bloodstream.

This, however, requires further investigation, especially if the relationship could change in dyslipidemia, and this should be achievable through lipidomics.

Existing evidence arising from lipidomic analysis of meibum samples from patients with chronic blepharitis, MGD and dry eye suggest that the biochemical and biophysical properties of meibum can change [33,76].

This includes specific alteration in polar versus non-polar lipids and changes in lipid orders due to increased saturation. Meibum cholesteryl esters are decreased whereas proteins are increased in MGD eyes, compared to normal [73]. The combination of lipid saturation and lipid–protein interaction is thought to make meibum less fluid (more viscous and more ordered), making it difficult for meibum to be delivered at the lid margin and further blocking the gland orifices [1,77]. The stasis of meibum can change the MG and ocular surface microbiome with a shift towards glands that produce lipases to degrade meibum and/or those that produce toxic mediators and some inflammation. Thus, this creates a vicious cycle of MG hyper-keratinization and compositional disturbance of meibum which could lead to self-perpetuating MGD [1,78]. However, MG (ductal) hyper-keratinization has recently been shown to be absent in a mouse model of age-related MGD [79], and this may be due to differences in models as there are in human subtypes of disease. Hence, hyper-keratinization may partly depend on the MGD phenotype and species-specific responses.

While the above studies underscore the usefulness of lipidomics in understanding the MG and secretion in the normal, primary MGD or dry eye situation, little has been done with lipidomics in elucidating the link between MGD and dyslipidemia [22,23,24,31,32]. We do not know for example if elevated levels of blood triglycerides, cholesterol, cholesterol esters or lipoproteins during dyslipidemia are accompanied by commensurate increases in the proportions of these lipid types in meibum. Further, we do not know if such a change could initiate MGD by disturbing meibum rheology, obstructing the MG and leading to eventual keratinization and atrophy of the gland [22,23,24,31,32]. It also remains to be determined if controlling dyslipidemia, in other words restoring the blood lipid profile to normal through diet or statin intervention, has any direct benefits to the MG.

Regarding dyslipidemia and MGD, the suggestion is that in using the DIO mouse model, we can, through lipidomics, characterize blood lipids and meibum lipids to determine if changing blood profiles have negative effects on meibum composition [22,23,24,31,34,36]. This could include evaluation of the relative abundances of the various lipid types (e.g., cholesteryl species) in both blood and meibum to ascertain if there is a direct or inverse relationship in terms of their abundances (increase or decrease) in blood and meibum. Additionally, we will be able to ascertain through lipidomics if dyslipidemia causes a shift in meibum saturation, increases meibum protein to lipid ratio, or raises meibum lipid orders, leading to impaired meibum fluidity [76,80]. These findings will be complemented with imaging of the mouse MG using meibography (Figure 3) to ascertain if the dyslipidemic mice demonstrate MG structural changes [1].

It is feasible to study the relationship between dyslipidemia and MGD in the DIO mouse model because the mice develop dyslipidemia in a natural fashion and over a short period time. Further, since the high-fat diet is used to induce dyslipidemia, the model offers an opportunity for a diet-based intervention, where dyslipidemia could be controlled by switching from the HFD to normal diet (ND). In this regard, it is also possible to study through lipidomic analysis if diet reversal (HFD to ND) normalizes the blood lipid profile and whether this intervention restores MG structure and function, i.e., meibum secretions. The findings could prove to have a translational benefit to the human condition, as they may inform new therapies for treating MGD and the related dry eye complications through dietary modification.

## 5. Conclusions

The role of dyslipidemia in the pathogenesis of MGD is not fully understood. This has been due, in part, to a lack of suitable animal models and appropriate techniques for investigating this relationship between dyslipidemia and the meibomian gland. Fortunately, the advent of lipidomics and a well-established mouse model of dyslipidemia provide a promising first step to better understand the effects of dyslipidemia on the meibomian gland, at the basic or experimental level.

## Figures and Tables

**Figure 1 ijms-20-03505-f001:**
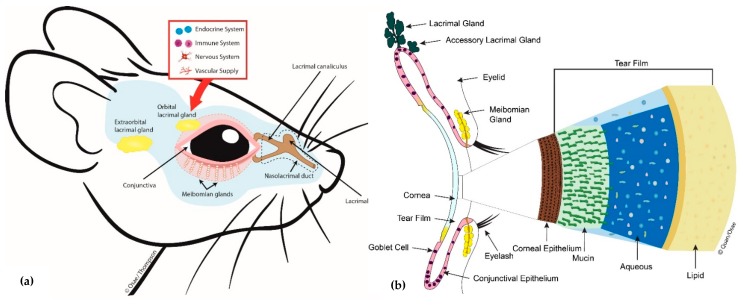
Diagram showing (**a**) mouse meibomian glands (lower eyelid only) and related ocular surface structures and (**b**) a cross-sectional view of some key lacrimal functional unit (LFU) components, the tear film components including the outermost lipid produced by the meibomian gland (MG) and how it interacts with the corneal surface.

**Figure 2 ijms-20-03505-f002:**
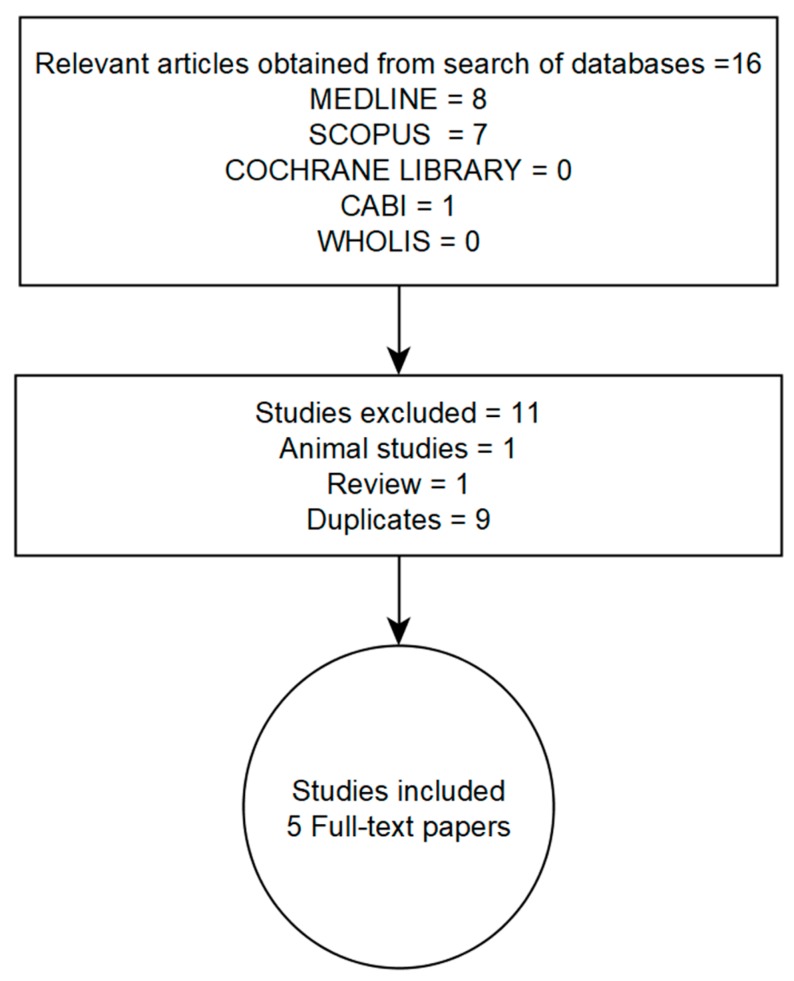
Flowchart showing databases searched and the selection and inclusion of the studies in this review. CABI = Center for Agriculture and Bioscience International, WHOLIS = World Health Organization Library Information System (WHOLIS).

**Figure 3 ijms-20-03505-f003:**
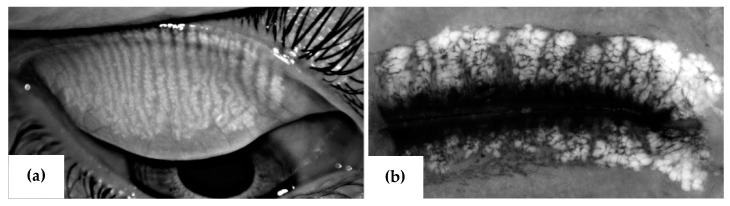
(**a**) Human (upper eyelid) meibomian glands acquired with Oculus® keratograph and (**b**) Mouse (upper and lower eyelids) meibomian glands acquired with a stereo microscope.

**Figure 4 ijms-20-03505-f004:**
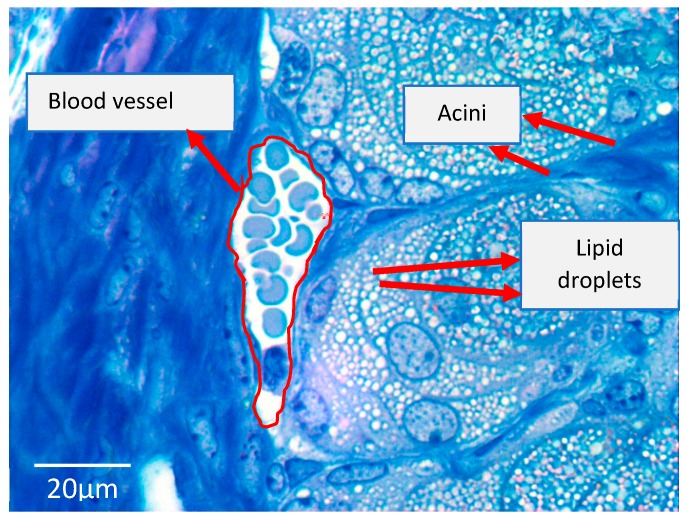
Light micrograph of an 8-week-old mouse meibomian gland stained with toluidine blue. A large blood vessel (red outline) containing erythrocytes is positioned close to the meibomian gland acini. Mag.100x.

**Table 1 ijms-20-03505-t001:** Summary of information from the included studies.

Study	Country/Ethnicity	Study Description	No. of Participants	Age Range (yrs)	No. of Meibomian Gland Dysfunction (MGD) Cases vs. Non-MGD	^†^ Prevalence Ratio of Dyslipidemia MGD: non-MGD	Named Serum Lipids Showing Significant Associations with MGD Status
Guliani et al. [31] (2018)	India/Indian	Prospective observational case-control hospital-based study	180	18–54	90 90	1.6	TC > 200 mg/dL TGs >150 mg/dL LDL > 130 mg/dL
Braich et al. [22] (2016)	Indian/Indian	Case-control hospital-based study	224	19–75	109 115	8.7	HDL < 40 mg/dL TC > 200 mg/dL TGs >150 mg/dL LDL > 130 mg/dL
Pinna et al. [32] (2013)	Italy/Italian	Observational, case-control pilot study	123	18–54	60 63	9.2	TC > 200 mg/dL TGs >150 mg/dL LDL > 130 mg/dL HDL > 40 mg/dL
Bukhari [23] (2013)	Saudi Arabia/N.S.	Prospective cohort study	236	15–78	132 104	~1.0	None
Dao et al. [24] (2010)	United States/N.S.	Retrospective case-control study	46 (cases only)	27–82	* 46 * Unknown no. of controls	** 1.5 ** Assuming n = 46 for controls	No specific tests of associations performed

^†^ Prevalence ratios were computed as: $\frac{\left(no.\;of\;MGD\;cases\;with\;dyslipidemia\right)\;}{\left(total\;no.of\;MGD\;cases\right)}/\frac{\left(no.\;of\;non-MGD\;cases\;with\;dyslipidemia\right)}{\left(total\;no.\;of\;non-MGD\;cases\right)}$. In computing prevalence ratios, TC > 200 mg/dL was considered as dyslipidemia for all studies except Bukhari et al.’s [23], where overall prevalence values of dyslipidemia among cases and control were used. TC = Total cholesterol, TGs = Triglycerides, LDL = Low-density lipoproteins, HDL = High-density lipoproteins, N.S. = Not specified.

**Table 2 ijms-20-03505-t002:** Modified Newcastle–Ottawa scale for assessment of the quality of included studies.

Quality Assessment Criteria	Acceptable Criteria	Guliani et al. [31] (2018)	Braich et al. [22] (2016)	Pinna et al. [32] (2013)	Bukhari [23] (2013)	Dao et al. [24] (2010)
**Selection**						
Definition of case or exposure (i.e., dyslipidemia and MGD status)	Adequate i. For dyslipidemia if based on at least an assessment of fasting triglycerides, total cholesterol, HDL or LDL levels	**✓**	**✓**	**✓**	**✓**	**✓**
	ii. For MGD if based on at least a symptom assessment, meibography, meibum expressibility or quality or slit lamp examination of morphologic eye lid features					
Representativeness of cases or exposed cohort?	Representative of average adult in community (age/sex/being at risk)	**✓**	**✓**	**✓**	**✓**	**✓**
Selection of controls or non-exposed cohorts	Specified as drawn from same the same community as cases or exposed cohort	-	**✓**	**✓**	**✓**	-
**Comparability**						
Study has sufficiently controlled for age/sex	Yes	**✓**	**✓**	**✓**	**✓**	**✓**
Study considered at least three additional risk factors for MGD	Aging, prolonged contact lens wear, recurrent eyelid infections, autoimmune disease e.g., Sjogren’s syndrome, Stevens–Johnson syndrome, use of certain drugs like isotretinoin, antihistamines, hormone replacement therapy	**✓**	**✓**	**✓**	-	**✓**
**Outcome**						
Assessment of outcome	Independent blind assessment, record linkage or self-report	-	-	-	-	**✓**
Response rate	Similar across groups	**✓**	**✓**	**✓**	**✓**	N/A
Same ascertainment method for cases and controls	Yes	**✓**	**✓**	**✓**	**✓**	-
Conclusively reports a direct link between MGD and dyslipidemia	Yes	-	-	-	-	-
**Overall quality score (Maximum 9)**		6	7	7	6	5

Each check (**✓**) mark equals one point and denotes a fulfilled criterion in the subsection. N/A = not applicable.

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
