# Peer review of "Dyslipidemia and Meibomian Gland Dysfunction: Utility of Lipidomics and Experimental Prospects with a Diet-Induced Obesity Mouse Model"

_ijms, 2019, doi:10.3390/ijms20143505_

Round 1
Reviewer 1 Report
This is the interesting review article regarding the relation between dyslipidemia and meibomian gland dysfunction. As meibomian glands are lipid producing glands, it been great concern if serum lipids are taken up and used for lipid synthesis in the acinar cells in the meibomian glands. The authors selected 5 papers for reviewing.
Although there's an established MGD definition by TFOS, there is not common diagnostic criteria throughout the world. For prevalence study, the diagnostic criteria is the key. Therefore, the authors should explain details about the diagnostic criteria of each manuscript.
And, because there mainly two types of MGD (hyper-/hypo-secretory MGD), the authors' DIO mice show hyper-secretory MGD? Description about the MGD type would be needed.
Author Response
Reviewer 1 : Comments and Suggestions for Authors
This is the interesting review article regarding the relation between dyslipidemia and meibomian gland dysfunction. As meibomian glands are lipid producing glands, it been great concern if serum lipids are taken up and used for lipid synthesis in the acinar cells in the meibomian glands. The authors selected 5 papers for reviewing. Although there's an established MGD definition by TFOS, there is not common diagnostic criteria throughout the world. For prevalence study, the diagnostic criteria is the key. Therefore, the authors should explain details about the diagnostic criteria of each manuscript.
RESPONSE: We are grateful for this comment. A statement on the diagnosis of MGD in different studies was made in the results section (SECTION 3.0) of the previous version of this manuscript. In this revised version, we have expanded the statement on the diagnosis and /or staging of MGD in the results section and included a full description per study in Appendix A.
And, because there mainly two types of MGD (hyper-/hypo-secretory MGD), the authors' DIO mice show hyper-secretory MGD? Description about the MGD type would be needed.
RESPONSE: We acknowledge that this was an important subject we missed in the earlier submission. We have included a statement on the type of MGD suspected in this model, and cited appropriately under section 4.2
Reviewer 2 Report
This manuscript was well-written. This topic seems interesting and helpful for readers.
Author Response
Reviewer 2: Comments and Suggestions for Authors
This manuscript was well-written. This topic seems interesting and helpful for readers.
RESPONSE: We are grateful for the kind comment.
Reviewer 3 Report
The review by Osae et al. discusses the potential link between dyslipidemia and meibomian gland dysfunction (MGD), the most common cause of dry eye disease. For example, they highlight a paper in 2016 that states ‘the odds of having dyslipidemia were 18 times greater among MGD patients than controls’, however, another paper in 2013 indicated no link between dyslipidemia and MGD. Therefore, the authors introduce a mouse model that they propose to use to study the link between dyslipidemia with MGD in the future. Overall, this is an interesting introduction to the potential links between dyslipidemia and MGD, I have only a few minor suggestions that may help improve the manuscript.
1/ Figure 1 is a nice depiction of the mouse lacrimal function unit, however, the arrow showing the cornea is technically pointing out the pupil, so I suggest moving this arrow to avoid any confusion. To fully illustrate the lacrimal functional unit here, I would suggest a cross-section of the cornea, and the tear film, alongside this figure to accurately highlight how meibum coats the aqueous layer and maintains hydration of the cornea.
2/ While dyslipidemia may, or may not significantly contribute to MGD, I think desiccative stress caused by low humidity environments (Suhalim et al., Ocular Surface 2014), and androgen depletion (Tamer C et al., Ophthalmic Res. 2006 ), should also be highlighted more as potential factors that contribute to MGD as they have received a lot of research focus in MGD studies in recent years. This is briefly mentioned in the discussion, but it may be more appropriate for the introduction. The authors also discuss ‘creating a vicious cycle of MG hyper-keratinization….’, however, recent evidence has shown an absence of hyper-keratinization in a mouse model of age-related MGD (Parfitt GJ., Aging 2013).
3/ I think listing the most up-to-date known lipidome of human and/or mouse meibomian glands is important here, as dyslipidemia causing an imbalance of the normal MG lipidome is the central hypothesis of this review. This is because I expect future lipidomic analysis in the mouse model of dyslipidemia to be performed in comparison with the known mouse MG lipidome. For example, in Brown, IOVS 2013 they state that the human lipidome is 52% wax esters, 44% cholesterol esters, 3.1% (O-acyl)-ω-hydroxy fatty acids, 1.5% triacylglycerol, 0.006% phospholipids. Is there newer literature available that states the known human or mouse MG lipidome?
4/ The authors point out that ultrastructural examination of the MG and its lipids is important to understanding MGD in the mouse model of dyslipidemia. They suggest imaging techniques such as meibography and electron microscopy, but I think it is also important to point out the powerful approach of Coherent anti-Stokes Raman spectroscopy (CARS) to probe the lipid:protein profile of meibomian glands (Suhalim et al., Ocular Surface 2014). This approach could be used to correlate dyslipidemia and lipidomics in human MGD samples and in the proposed mouse model.
Author Response
Reviewer 3: Comments and Suggestions for Authors
The review by Osae et al. discusses the potential link between dyslipidemia and meibomian gland dysfunction (MGD), the most common cause of dry eye disease. For example, they highlight a paper in 2016 that states ‘the odds of having dyslipidemia were 18 times greater among MGD patients than controls’, however, another paper in 2013 indicated no link between dyslipidemia and MGD. Therefore, the authors introduce a mouse model that they propose to use to study the link between dyslipidemia with MGD in the future. Overall, this is an interesting introduction to the potential links between dyslipidemia and MGD, I have only a few minor suggestions that may help improve the manuscript.
1/ Figure 1 is a nice depiction of the mouse lacrimal function unit, however, the arrow showing the cornea is technically pointing out the pupil, so I suggest moving this arrow to avoid any confusion. To fully illustrate the lacrimal functional unit here, I would suggest a cross-section of the cornea, and the tear film, alongside this figure to accurately highlight how meibum coats the aqueous layer and maintains hydration of the cornea.
RESPONSE: We sincerely thank the reviewer for the great suggestions made to improve the figure. Figure 1( now fig 1a) has been edited and a cross-sectional view showing the LFU in relation to the tear film and its lipid produced by the meibomian glands have been included as fig.1b to show how it interacts with the cornea.
2/ While dyslipidemia may, or may not significantly contribute to MGD, I think desiccative stress caused by low humidity environments (Suhalim et al., Ocular Surface 2014), and androgen depletion (Tamer C et al., Ophthalmic Res. 2006 ), should also be highlighted more as potential factors that contribute to MGD as they have received a lot of research focus in MGD studies in recent years. This is briefly mentioned in the discussion, but it may be more appropriate for the introduction. The authors also discuss ‘creating a vicious cycle of MG hyper-keratinization….’, however, recent evidence has shown an absence of hyper-keratinization in a mouse model of age-related MGD (Parfitt GJ., Aging 2013).
RESPONSE: We have highlighted the role of androgen insufficiencies in the development of MGD and cited appropriately under section 1. We have also distinguished the presence and absence of MG hyper-keratinization in different MGD phenotypes and cited appropriately under section 4.3
3/ I think listing the most up-to-date known lipidome of human and/or mouse meibomian glands is important here, as dyslipidemia causing an imbalance of the normal MG lipidome is the central hypothesis of this review. This is because I expect future lipidomic analysis in the mouse model of dyslipidemia to be performed in comparison with the known mouse MG lipidome. For example, in Brown, IOVS 2013 they state that the human lipidome is 52% wax esters, 44% cholesterol esters, 3.1% (O-acyl)-ω-hydroxy fatty acids, 1.5% triacylglycerol, 0.006% phospholipids. Is there newer literature available that states the known human or mouse MG lipidome?
RESPONSE: We have provided a list of major and “minor” meibum lipids under section 4.2. We acknowledge that Brown et al., have the most recent (2013) list of lipids in human meibum but their work unlike that of Butovich et al., (2012) does not include a list of meibum lipids in animals. Therefore, we considered that of Butovich et al., as one that allows for adequate vis-à-vis comparison of human and meibum lipids. However, much work is still needed to elucidate the full lipidome of human and animal meibum.
4/ The authors point out that ultrastructural examination of the MG and its lipids is important to understanding MGD in the mouse model of dyslipidemia. They suggest imaging techniques such as meibography and electron microscopy, but I think it is also important to point out the powerful approach of Coherent anti-Stokes Raman spectroscopy (CARS) to probe the lipid:protein profile of meibomian glands (Suhalim et al., Ocular Surface 2014). This approach could be used to correlate dyslipidemia and lipidomics in human MGD samples and in the proposed mouse model.
RESPONSE : We thank reviewer for suggesting the utility of CARS technology. We have mentioned its usefulness in investigating MG changes in the presence of dyslipidemia and have duly cited references to support this under section 1 .